# Assessing Lifestyle in a Large Cohort of Undergraduate Students: Significance of Stress, Exercise and Nutrition

**DOI:** 10.3390/nu16244339

**Published:** 2024-12-16

**Authors:** Daniela Lucini, Ester Luconi, Luca Giovanelli, Giuseppe Marano, Giuseppina Bernardelli, Riccardo Guidetti, Eugenio Morello, Stefano Cribellati, Marina Marzia Brambilla, Elia Mario Biganzoli

**Affiliations:** 1BIOMETRA Department, University of Milan, 20129 Milan, Italy; luca.giovanelli@unimi.it; 2Exercise Medicine Unit, Istituto Auxologico Italiano IRCCS, 20135 Milan, Italy; g.bernardelli@unimi.it; 3Department of Biomedical Sciences for Health, University of Milan, 20157 Milan, Italy; ester.luconi@unimi.it; 4Medical Statistics Unit, Department of Biomedical and Clinical Sciences L. Sacco, “Luigi Sacco” University Hospital, University of Milan, 20157 Milan, Italy; giuseppe.marano@unimi.it (G.M.); elia.biganzoli@unimi.it (E.M.B.); 5Data Science Research Center, University of Milan, 20157 Milan, Italy; 6DISCCO Department, University of Milan, 20122 Milan, Italy; 7Dipartimento di Scienze Agrarie e Ambientali—Produzione, Territorio, Agroenergia—University of Milan, 20133 Milan, Italy; riccardo.guidetti@unimi.it; 8Laboratorio di Simulazione Urbana Fausto Curti, Dipartimento di Architettura e Studi Urbani, Polytechnic of Milan, 20133 Milan, Italy; eugenio.morello@polimi.it; 9SEGE Srl, 20146 Milan, Italy; stefano.cribellati@se-ge.com; 10Department of Language Mediation Sciences and Intercultural Studies, University of Milan, 20122 Milan, Italy; marina.brambilla@unimi.it

**Keywords:** well-being, physical activity, tailored intervention, stress management, public health, lifestyle assessment, physical activity, nutrition quality

## Abstract

Background/Objectives: Lifestyle (in particular, nutrition and exercise) determines present and future youths’ health. The goal of the present study was to identify specific student groups who deserve precise lifestyle improvement interventions, tailored to their characteristics. Methods: An anonymous web-based questionnaire to assess lifestyle was posted on the websites of two main Italian Academic Institutions, and 9423 students voluntarily participated. A personalised immediate report was provided to improve compliance/motivation. We assessed age, sex, affiliation, anthropometrics, lifestyle components (nutrition, exercise, sedentariness, stress perception, smoking, alcohol, sleep), and the desire to be helped with lifestyle improvement. Cluster analysis was performed to identify healthy lifestyle groups among the students. Results: In total, 6976 subjects [age: 21 (20, 23) yrs; 3665 female, 3300 male] completed the questionnaire and were included. Of these students, 73.9% expressed the need for lifestyle improvement help, particularly for becoming physically active (66.7%), managing stress (58.7%), and improving nutrition (52.7%). We unveil three clusters of subjects, each corresponding to a distinct lifestyle pattern. The clusters are differentiated by exercise level and perceptions of stress/fatigue/somatic symptoms (cluster 1: 74.8% meet international exercise guidelines (IEGs), 67.4% have high stress perception, 49.1% drink 1–3 glasses of wine/beer per week, and 63.3% drink 0–1 glass of spirits per week; cluster 2: 75.6% meet IEGs, 75.7% have low/medium levels of stress perception, and 65.8% have low alcohol consumption; cluster 3: 72.5% do not meet IEGs, 77.6% have high stress perception, and 67.5% have low alcohol consumption). More active students present lower stress/somatic symptoms perception. Interestingly, the AHA diet score (nutrition quality) was not in the ideal range in any cluster (nevertheless, obesity was not of concern), being worst in cluster 3, characterized by higher stress perception (59.7% had poor nutrition quality). Those who were physically active but showed a high stress/fatigue perception were used to drinking alcohol. Conclusions: Students desire help to improve their lifestyle, and this approach might help identify specific student groups to whom LIs in Academic Institutions can be tailored to foster well-being and promote health.

## 1. Introduction

Undergraduate students (approximately 18–26 years) are typically in the phase of young adulthood, a pivotal time of life. They are different from older adults and adolescents in ways that affect their decision-making, behaviour, and health choices [1]. During these years, they complete their education and pursue endeavours which will shape their adult personal and working life. They also shape their attitude towards behaviours which may dramatically impact their well-being and health in the present and future, and the scientific literature [2] depicts this age as one of the best periods in the lifespan for health promotion and the primary prevention of chronic non-communicable diseases. Nevertheless, motivating young people to adopt a healthy lifestyle (defined as individual behaviours having a significant impact on health and well-being, such as nutrition, physical activity, stress management, sleep hygiene, and consuming risky substances, for example smoke and alcohol) is demanding since they frequently exhibit health behaviours below public health recommendations [3,4]. A recent study on a large Spanish student population [5] conducted during the COVID-19 pandemic revealed that a lack of time and laziness were indicated as the main reasons for giving up or not taking up physical activity. Other recent studies addressing nutrition in students have shown that a “healthy diet” pattern is present only in subjects who exercise on a regular basis [6], that there is evidence of a correlation between poor diet quality in students with elevated BMI and smoking, stress level, alcohol consumption, and poorer sleep quality [7], and that there is an association between healthy diet and physical activity [8].

Generally, young people consider themselves healthy individuals and are more prone to focusing “on the present” instead of “on the future”, at least regarding health issues. To promote healthy behaviours among young populations, it might be particularly interesting to draw their attention to the “immediate” benefits of a healthy lifestyle; among these benefits, the perception of well-being and improved quality of life may play a pivotal role [9]. It might be particularly appropriate to change young adults’ “point of view” [10,11,12] via preventive strategies, focusing more on well-being and the promotion of healthy behaviours (in particular, a physically active life, healthy nutrition, non-smoking, stress management, and sleep hygiene) than solely on reducing traditional cardio-metabolic risk factors (such as high cholesterol level, high blood pressure, overweight/obesity, etc), which often are within the normal range. Recently, we have shown [9] that young employees present a worse lifestyle than older ones, even without alterations in anthropometric, metabolic, lipid, and haemodynamic parameters. Unveiling unhealthy behaviours (for instance, poor quality of nutrition, smoking, or sedentariness) before the appearance of abnormal clinical parameters (such as those related to metabolic syndrome and low-grade chronic inflammation, such as hyperglycaemia, obesity, and high cholesterol level) may help young people to reconsider their habits and modify them as soon as possible.

It is important to underline that the worsening of psychological well-being linked to anxiety and stress is particularly interesting in youth. Stress and depression represent an emerging health issue [13,14], especially in young populations. They live in a rapidly changing and demanding world, where individual and global sources of stress contribute to the feeling that continually increasing performance is a necessity, which may not be counterbalanced by enough individual and community coping resources. The link between stress and lifestyle is complex, and many pieces of evidence suggest that stress may worsen lifestyle [11,14,15], with those affected favouring poor nutrition, sedentariness, smoking, alcohol, drug abuse, etc., as coping strategies, with a negative consequence on cardiometabolic–oncological health. On the other hand, lifestyle improvement has been shown to be a pivotal strategy to prevent chronic diseases, improve well-being, and manage stress [11,14,15,16,17]. Becoming/remaining physically active plays a central role in this context [11,16]. Many Academic Institutions [18,19] offer their students services to promote and improve well-being, including psychological support and lifestyle improvements, and some scientific papers have been devoted to the study of students’ lifestyles. For instance, one study [20] showed, in a large cohort of Brazilian students, that the odds of depression and anxiety symptoms were higher in students characterized by sedentary behaviour; another study [21] revealed in German students that lower sedentary time and higher physical activity were associated with reduced levels of perceived stress. Another one [22], using cluster analysis, showed that students that smoked were more likely to report higher stress.

Considering the great importance of fostering health and well-being in young people, our hypothesis was that tailored approaches would be more effective in improving lifestyles than “generic” intervention. The present study aimed to define groups of students characterized by specific lifestyles to better tailor preventive strategies and educational procedures to favour well-being and health during academic years. To this end, we used data collected by means of a web-based anonymous questionnaire [11,23,24], filled in by a large cohort of undergraduate students from the two largest universities in the northern area of Italy (Lombardy), to investigate lifestyle components, with particular regard to nutrition habits, physical activity, perception of stress, fatigue, somatic-stress-related symptoms, the consumption of alcoholic beverages, smoking habits, and sleep.

## 2. Materials and Methods

### 2.1. Participants

This study is part of a project of the University of Milano and the Polytechnic of Milano; the goal is to put in place some best practices to improve quality of life, well-being, and sustainable lifestyle in the city area where the two big Academic Institutions are located. The promotion of healthy lifestyle behaviours has a role of paramount importance and may be considered a real sustainable tool. To take action today (to improve individual behaviour) is to preserve a great good (health) that, otherwise, might disappear [25,26]. This specific topic was ideated through a multidisciplinary collaboration between the Head and members of the residency program of Sports Medicine and Physical Exercise, several professors and experts in various areas of personal health and well-being from the University of Milan, and the Governance and University Administration. A web-based questionnaire, completely anonymous, was posted on the website of the two Academic Institutions in 2019. All the students of the Polytechnic, students of select courses at the University of Milan (Medicine and Surgery, Medical Biotechnologies, Medical Biotechnology and Molecular Medicine, Nursing, Audiometric Techniques, Viticulture and Enology, Foodservice Science and Technology, Exercise Sport and Health Sciences, Philosophy, Philosophic Sciences, International Studies and European Institutions, and Political Sciences), and all students enrolled in their first year of any course were invited to fill it out. An email explaining the purposes and goals of the questionnaire and the possibility of disseminating the results derived from the anonymous analysis was sent to the students. A personalized immediate report was provided to improve compliance/motivation. We have already described [23] the methodology employed to create the questionnaire. During the pandemic period, we offered students online healthy lifestyle promotion programs [26].

### 2.2. Instruments and Procedure

We collected anthropometric data (weight, height, waist circumference), age, sex, and university affiliation.

Lifestyle Assessment:

Physical activity (total activity volume) was assessed by a modified version of the commonly employed short version of the International Physical Activity Questionnaire (IPAQ) [27], which focuses on the intensity (nominally estimated in Metabolic Equivalents—METs—according to the type of activity) and duration (in minutes) of physical activity. We considered the following levels: activities of moderate intensity (≈4.0 METs/minute) and activities of vigorous intensity (≈8.0 METs/minute). These levels were used to calculate the weekly exercise volume of structured exercise (METsMV; MV—moderate and vigorous) using the following equations:METsMV = 4 × M × dM + 8 × V × dV,(1)
where METsMV stands for weekly moderate and vigorous physical activity volume expressed in METs minutes/week; M is the number of minutes/day of moderate-intensity activity; dM is the number of days/week of moderate-intensity activity; V is the number of minutes/day of vigorous-intensity activity; and dV is the number of days/week of vigorous-intensity activities. The quantity in (1) may then be considered the total weekly volume of structured exercise.

We also assessed the frequency of regular strength and flexibility exercises, considering the following scale: never; sometimes; 1 session/week; 2–3 sessions/week; more than 3 sessions/week.

Sedentary behaviour was assessed by asking the number of hours spent in sedentary behaviour (for instance, studying, sitting, driving, TV viewing, computer or smart device usage) during weekly working days and weekend days.

Nutrition was assessed using the American Heart Association (AHA) Diet Score [28], taking into consideration fruit/vegetable, fish, sweetened beverage, whole grain, and sodium consumption (the assessment of the latter was adapted to Italian eating habits and considered as a score of “nutrition quality”) [23].

Perceptions of stress, fatigue, and subjective somatic-stress-related symptoms (short 4SQ) were assessed using a self-administered questionnaire [23] with nominal self-rated Likert scales from 0 (“no perception”) to 10 (“highest perception”) for each measure. We considered a short version of the 4SQ, taking into account 3 somatic symptoms (perception of heart beating, perception of muscle tension, perception of knot in stomach); thus, the total score ranged from 0 to 30.

Smoking behaviour: We considered all subjects who reported to have never smoked or to have stopped smoking for more than one year as non-smokers.

We enquired about the usage of alcohol, considering Italian habits, asking the number of glasses of wine or beer consumed per week and the number of glasses of spirits consumed per week.

Perceptions of quality of sleep, quality of health, and quality of life were assessed with nominal self-rated Likert scales from 0 (“bad”) to 10 (“very good”) for each measure.

Desire to be helped with lifestyle change was inquired considering two options: yes or no.

### 2.3. Data Analysis

A quality check of the collected data was performed to remove non-realistic answers from the dataset and to identify conditions with a high percentage of non-response. All participants voluntarily provided anonymous data. The study protocol was approved by the Institutional Ethics Committee of the University of Milan (Allegato 4 Comitato Etico 25.05.18; Repertorio pareri Comitato Etico: parere numero 21/18) and by the Institutional Ethics Committee of the Polytechnic of Milan (Parere 11/2018 dated 26 July 2018).

Statistics: The data consisted of records of 9423 students who filled out the questionnaire; 2447 students who had more than five missing values were excluded. Categorical variables were summarized by counts and percentages, and numerical variables were summarized using the median and the first and third quartiles due to the asymmetry of the distributions. Comparisons between genders were carried out using quantile regression methods for numerical variables and logistic regression for categorical variables with binary or multiple response options. *p*-values were corrected for the multiplicity of comparisons using the Bonferroni rule. The primary aim was to identify healthy lifestyle groups among the students; to this aim, cluster analysis was performed. The following variables were used: waist circumference, AHA diet score, METs for moderate and vigorous activities, BMI, sedentary time during week days, sedentary time during weekends, hours of sleep, smoking habits, perceptions of stress, fatigue, and subjective somatic-stress-related symptoms (s4SQ), consumption of wine and beer, and consumption of spirits. All the numerical variables for the above were converted into categorical variables using the following classifications for the cluster analysis:Waist circumference (WC) was coded as green (<80 cm and <94 cm, respectively, for female and male students), yellow (80–87.9 cm and 94–100.9 cm for female and male students, respectively), or red (>87.9 cm and >101.9 cm, respectively) [29]. Students who declared WC < 60 cm or >130 (female) and WC < 70.5 cm or >130 (male) were excluded from the analysis.BMI (body mass index) was coded as underweight/normal weight (<25 Kg/m^2^), overweight (25–29.9 Kg/m^2^), or obese (30 Kg/m^2^) [28].METs for moderate and vigorous activity were codified as “insufficient levels of exercise” if <600 (MET·min/week), or otherwise as “adequate” [27].Weekly hours spent in sedentary activity (during working days and weekends) were coded as “active habits” (<9 h/week) or “sedentary habits” (otherwise) [30].Hours of sleep were considered “adequate” if ≥7 per day or “insufficient” otherwise [31].Consumption of wine and beer was coded as 0 glasses/week, 0.1–1 glasses/week, 1.1–2.9 glasses/week, or >2.9 glasses/week.Consumption of spirits: 0 glasses/week, 0.1–1 glasses/week, or >1 glasses/week.Perception of subjective stress-related somatic symptoms (s4SQ) was coded into five classes in the following quintiles: 0–3, 4–6, 7–10, 11–15, and 16–30.Perceptions of stress and fatigue were coded, separately, into the following five classes: 0–2, 3–4, 5–6, 7–8, and 9–10.Smoking habits were coded as “smoker, ex-smoker, electronic cigarettes, or non-smoker”.

Eleven subjects who declared “other gender” were excluded due to the impossibility of obtaining the classes for waist circumference, which considered only values for male and female categories. Cluster analysis was performed on 5861 subjects with complete records of the variables above. In a preliminary step, the association among the variables above was evaluated by multiple correspondence analysis [32]. The clustering algorithm used was K-modes [33] because it is suited for categorical variables. In addition, it is optimal for survey research applications because it can handle large datasets with a high number of categories of variables. The algorithm was run many times (6000 runs for k = 3 to 10 clusters), and the optimal number of clusters was chosen according to the maximum of the average silhouette width index [34]. Clusters were described using graphical representation (heatmap) and textual description. To investigate the degree of separateness of the clusters, we applied Principal Coordinates Analysis (PCoA) methods, as described by Bakker [35]. The analysis was performed using R software version 4.0.4 [36] with the packages FactoMineR [37], KlaR [38], and vegan [39] added.

## 3. Results

The questionnaire was filled out by 9423 subjects. Of these, 7036 were Polytechnic students (74.6%), 2205 were students of the University of Milan (23.4%), and 182 (2.0%) did not specify affiliation. Based on the pre-defined criteria (see statistical analysis), 6976 (74.03%) subjects were included in the analysis. The features of these students are summarized in Table 1.

Table 1 shows that the majority of students are normoweight, though the percentages of male students in the overweight and obesity classes are slightly greater than those of female students. Male students are slightly more active than female students who, instead, present a higher perception of stress, fatigue, and somatic symptoms. The majority of students are non-smokers and are occasional alcohol consumers, with the percentage of subjects who do not drink any alcohol being slightly greater in female students.

Notably, the majority of students were characterized by a normal BMI and normal waist circumference classes (“green”). Nevertheless, the median of the AHA diet score corresponded to “intermediate health” [28], suggesting a quality of nutrition deemed in need of improvement. Moreover, while most students reported meeting international physical activity guidelines [40] regarding endurance exercise, only a tiny percentage (more evident in male subjects) performed strength/flexibility exercises regularly (2/3 sessions/week), as per the guidelines. Notably, 78.1% of female and 69.3% of male students desired help with making a lifestyle change, particularly for improving exercise (66.7%), managing stress (58.7%), and improving nutrition (52.7%).

### Investigation of Lifestyle Patterns

The association between the 13 variables used for assessing students’ lifestyles (see Methods section for their definition) was evaluated via MCA (Appendix A). The first dimension (Dim. 1) and the second dimension (Dim. 2) together explained 78.3% of the variance. Dim.1 (*X*-axis) explained 61.4% of the total variability, and three variables showed high coordinates in this dimension (s4SQ, fatigue, and stress level), suggesting that these three variables are associated. Dim. 2 (*Y*-axis) explained 16.9%. Wine and beer, spirits, and (to a lesser extent) smoking showed a high coordinate on this axis, suggesting another association.

More details about the associations can be derived from Figure 1 (MCA map). There is an association between low fatigue scores, low stress scores, and low perceptions of somatic symptoms (s4SQ score), all represented on the right side of the plot. In contrast, higher levels of the same variables are all represented on the left side, indicating that they are associated with each other. Dim.1 (*X*-axis) separated subjects with a low perception of stress, fatigue, and somatic symptoms from subjects with a high level of these perceptions. At the same time, Dim.2 (*Y*-axis) separated subjects with a high consumption of wine/beer and spirits (and smokers) (observed in the top part) from subjects with low levels of the same variables (observed in the bottom part), suggesting that the consumption of alcohol and smoke are slightly associated with one another.

Subsequently, cluster analysis was performed. According to the average silhouette width, 5861 students were grouped into three clusters (Appendix A). Each cluster corresponds to a distinct lifestyle pattern. Figure 2 shows the characterization of clusters in terms of student’s features, and the corresponding textual description is reported in Figure 3. It may be seen that the major degree of association among the variables of interest (which emerged from the MCA) is reflected in the clusters’ composition. Subjects in clusters 1 and 3 present high stress and fatigue perception, along with medium–high perceptions of somatic stress-related symptoms; subjects in cluster 2 report low perceptions of somatic-stress-related symptoms and low perceptions of stress. Similarly, wine/beer and spirit consumption are connected in each of the three clusters. In the clusters, we do not find an association between smoking habits and alcohol consumption; however, this is likely due to the low degree of association.

From Figure 2, it may be seen that all the variables contribute to defining distinct lifestyle patterns, except five variables, which are smoking habit (in fact, in all three clusters, the highest frequency is “non-smoker”), waist circumference (“green” category common to all the clusters), body mass index (<25 Kg/m^2^ common to all the clusters), hours of sleep (“adequate” category common to all clusters), and sedentariness during weekends.

Figure 3 shows the three unveiled clusters of students. Students in cluster 1 are physically active, present a high perception of stress, and drink alcohol; 67.9 of them are characterized by an AHA diet score of 2–3, corresponding to “intermediate health” [28]. Cluster 2 is composed of students characterized by the best lifestyle pattern: being physically active, presenting a low stress perception, and not drinking alcohol. Nevertheless, 68.5% of them are characterized by an AHA diet score of 2–3, corresponding to “intermediate health” [28], suggesting a quality of nutrition deemed in need of improvement. Students in cluster 3 are sedentary, exercise less than recommended by international guidelines [40], and present a high perception of stress. Notably, 59.7% of them are characterized by an AHA diet score of 0–1, corresponding to “poor health” [28], suggesting a quality of nutrition deemed in need of great improvement.

Remarkably, more active students present lower stress perception; moreover, students in cluster 1, who are physically active but show a high perception of stress/fatigue, are used to drinking alcohol.

PCoA was performed to investigate the degree of separateness of the three clusters. Figure 4 depicts that the clusters show some degree of separateness, even though some overlapping occurs in the central part of the figure.

Concerning the relationships between lifestyle and “external” features, we note the following: In cluster 1, 56% (C.I. 53–58%) are female and 44% are male. In cluster 2, 47% are female and 53% (C.I. 51–56%) are male. In cluster 3, 65% are female and 35% are male. Moreover, the median of the judgement on quality of health is higher in cluster 2 (eight points) as compared to clusters 1 and 3 (seven points for each one), but the difference among the three groups was not statistically significant (Wald test: *p* = 0.38)

## 4. Discussion

We hypothesized that the possibility of defining specific student groups, characterized by different lifestyle patterns, might help to better outline tailored intervention programs to be offered to the undergraduate students of two main Academic Institutions in northern Italy. Therefore, cluster analysis was employed to identify groups of students with similar lifestyles to better plan interventions aimed at lifestyle change.

In this paper, by employing an anonymous web-based questionnaire and a large cohort of undergraduate students, we unveil three major clusters of subjects, each corresponding to a distinct lifestyle pattern. Clusters are differentiated particularly in relation to exercise behaviour and perceptions of stress, fatigue and somatic-stress-related symptoms; more active students present lower perceptions of stress and somatic symptoms. Interestingly, we observed that the AHA diet score (considered a marker of nutrition quality) was not in the ideal range in any cluster (nevertheless, obesity was not of concern in this sample), being worst in the cluster characterized by higher perceptions of stress; students in cluster 1, who were physically active but showed a high perception of stress and fatigue, were more used to drinking alcohol. Notably, a great percentage of students desired to be helped with lifestyle improvement, particularly in terms of becoming more physically active, managing stress, and improving nutrition.

Lifestyle represents an important tool to promote health, prevent/manage chronic non-communicable diseases [12,28,40,41], and improve prognosis even in communicable diseases such as COVID-19 [42]. In addition, it is of paramount importance to foster well-being and manage stress [43,44]. Thus, improving lifestyle choices is valuable and essential for everyone, but particularly young people. Unfortunately, young people are often characterized by poor lifestyles, reduced well-being, and increased stress perception [13,14,44]—conditions which drive them to ask for help. In this paper, we observed that at least 60% of the students who filled in the lifestyle questionnaire desired to be helped in managing stress, and at least three-quarters of them in improving their lifestyle, particularly in terms of becoming physically active and having healthier nutrition. This observation suggests that institutions, particularly academic ones, should have a critical role in this regard. Many of them guarantee their students services to foster well-being [18,45,46,47] and promote healthy lifestyles. The possibility to tailor interventions to specific groups may represent a critical approach toward an efficacious result. Simple generic counselling may not always drive a real behavioural change [48]; on the contrary, tailored, specifically designed interventions considering group or individual characteristics and needs may be beneficial [49]. In this study, we unveiled specific student clusters characterized by specific lifestyles, considering lifestyle data collected by means of a simple web-based questionnaire. Notably, students with high stress perceptions are the least active and have the worst nutrition quality (cluster 3), while students who present similarly high stress perceptions but are physically active (cluster 1) drink more alcohol. On the other hand, students of cluster 2, who have a healthier lifestyle, may deserve attention in improving the quality of their nutrition, which, while being better than in cluster 3, was not ideal (as recommended by guidelines) [28] (poor = score 0–1; intermediate = score 2–3; ideal = score 4–5). The observation that the quality of nutrition may be suboptimal in normal-weight subjects may merit a specific note, in light of the importance of addressing this specific issue in preventive campaigns or in well-being-promoting campaigns; as such, placing attention only on overweight and obesity could be misleading.

It may be particularly useful for young subjects to design stress management and/or stress prevention campaigns that also include interventions based on lifestyle improvement; in this context, exercise may play a pivotal role [50]. On the other hand, it is important to educate students about the different benefits/risks of exercise and sport. For instance, excessively encouraging the pursuit of only high-level sports performance might lead to abandoning sports (if reached performances are not considered of note), increased risk of injuries, and promoting risky behaviours such as alcohol use, especially in subjects who present high stress perception. Vice versa, fighting sedentariness might be important to prevent the adoption of this unhealthy behaviour, which is frequently associated with stress and might accompany the student for life. Also, campaigns aimed at promoting healthy nutrition may have a significant role in stress management interventions. The link between stress and overweight and obesity is well known; interestingly, in this paper, we observed that poor nutrition quality, stress perception, insufficient exercise, and sedentariness clustered together (cluster 3), even though obesity was not of concern in these subjects. Educating young people before the appearance of clear signs of disease, such as elevated body weight, could represent a meaningful intervention, particularly in students characterized by elevated stress perception, a condition which might favour malnutrition.

The possibility of unveiling different student clusters identified by diverse lifestyle characteristics might be useful to tailor interventions and avoid general campaigns that might be suboptimal in fostering lifestyle changes.

Assessing lifestyle in undergraduate students represents an important research topic due to the possible translational implications for intervention designs in the academic setting. Several researchers have addressed this topic in different countries [5,6,7,8,20,21,51,52,53,54], using many different approaches. These studies consider that students’ lifestyles are shaped by various factors, including financial opportunities [22], access to health resources, and social norms that influence their approach to health, physical activity, nutrition, stimulants, and mental health. To the best of our knowledge, most of these studies were descriptive surveys, showing the following: most of the students analysed were characterized by normal weight (with male students being slightly overweight [51,52,53]); male students are slightly more physically active [4] than female students; stress is of concern, particularly for female students [9,55,56]; and the majority of students are non-smokers and are occasional alcohol consumers [4,51,52,53]. Of particular interest is that, a more sophisticated approach, utilized in some of these papers, permitted the unveiling of important practical aspects, such as the relationship between depression and anxiety symptoms and sedentary behaviour [20]; the relationship between smoking habits and reported stress [22]; and the relationship between a healthy diet and regular exercise [6].

Notably, two major aspects were assessed in the literature: the association between physical activity and stress and the association between nutrition/diet patterns and physical activity. This issue assumes particular importance considering the strong roles of stress and unhealthy nutrition in worsening health/well-being and in many chronic diseases. On the other hand, regular aerobic exercise represents a pillar of the strategies to improve health and treat/prevent chronic disease, but also to help manage stress and nutrition patterns.

In the present study, we confirm some of the observations of other studies and add a new interesting aspect: the differences in lifestyle patterns among different clusters of students, corroborating the importance of tailoring interventions based on specific characteristics. We applied a multidimensional approach to the lifestyle assessment and cluster analysis, which allowed us to unveil that the main factors capable of distinguishing the different clusters of students were exercise, stress perception, nutrition quality, and alcohol. This approach allowed us to identify groups of students with distinct healthy behaviours and characterize them in terms of exercise, stress perception, nutrition quality, and alcohol consumption.

The numerosity of the study sample may also be of note. The majority of previous studies considered small student cohorts, except for few of them [8,20], while our study analysed data from thousands of subjects, and this numerosity allowed us to obtain reliable results.

Academic health promotion and health management may grant benefits which result in improvements in undergraduate students’ lifestyles; in fact, education efforts to promote healthy lifestyles may be disseminated to academic employees and the general population. Moreover, undergraduate students may serve as models in their present and future lives [57], when (presumably) they will fill an important professional role. Lastly, specific training on lifestyle approaches for students of courses directly involved in the management of diseases and health promotion (such as medical students, nurses, physiotherapists, exercise physiologists, etc.) could become mandatory [58].

The employed questionnaire had already been used in other campaigns outside academia [11,23,24] to define the lifestyles of employees [11,23] or patients. It has been found to be capable of revealing an association between active habits and low stress perception both in healthy subjects [23] and breast cancer survivors; of revealing the betterment of different lifestyle components after a period of physical training in metabolic syndrome patients [25] and in stress management interventions [59]; of showing that young employees are characterized by poor lifestyle compared to older employees [9]; and of demonstrating an association between stress perception and markers of sympathetic overactivity [25,59]. The possibility to easily quantify stress perception using just three questions inquiring about stress from both a cognitive (directly asking about stress perception: “Do you feel stressed?”) and somatic (asking questions regarding perceptions of fatigue and other somatic symptoms, such as palpitations or muscular tension) point of view may offer a simple metric for assessing and monitoring interventions [25,59].

Limitations: This study presents some limitations. In general, the data obtained through self-reported questionnaires might be of suboptimal quality. On the other hand, the high number of respondents and the quality of data analysis may help control this aspect [23]. Moreover, the questionnaire was completely anonymous, and we provided participants with a personalized, immediate report based on the filled information, hence increasing their likelihood of compliance [23] to insert trustful data on their present condition. The questionnaire was filled out voluntarily, although a sampling selection bias is expected, meaning that caution should be taken when extending the results to the overall student population of the two universities. Nevertheless, the number of respondents was relatively high for such an investigation, and we observed an extensive range of scores for all questions. Another limitation of our study should be acknowledged: we focus our attention only on the possibility of unveiling specific student clusters (using cluster analysis), possibly losing the opportunity to report on other results relevant to the community that would require different statistical approaches. Our large dataset could have permitted in-depth analysis. Nevertheless, we decided to focus our attention only on a specific goal: to define groups of students characterized by specific lifestyles to better tailor interventions aimed at improving health and well-being, answering a precise question of our Academic Institutions. Future lines of data analysis will definitely consider different statistical approaches, such as other types of parametric tests, the use of inferential statistics, etc., granting results that will be more generalizable to other contexts.

## 5. Conclusions

In conclusion, we report that students of the two main public universities in North Italy desire help with lifestyle improvement, mainly to be more physically active, to be capable of managing stress, and to have better nutrition. By using an ad hoc web-based lifestyle questionnaire, we revealed the presence of three main clusters of subjects characterised by different lifestyle patterns. Students who reported high perceptions of stress, fatigue, and somatic-stress-related symptoms were also less physically active and had the worst nutrition quality (cluster 3). Moreover, students with higher perceptions of stress but that were physically active (cluster 1) were those who consumed alcohol. However, students who were physically active and reported low perceptions of stress, fatigue, and somatic-stress-related symptoms (cluster 2) showed a need to improve nutrition quality. This approach might help identify specific student groups to tailor interventions fostering well-being and promoting health in academic settings.

## Figures and Tables

**Figure 1 nutrients-16-04339-f001:**
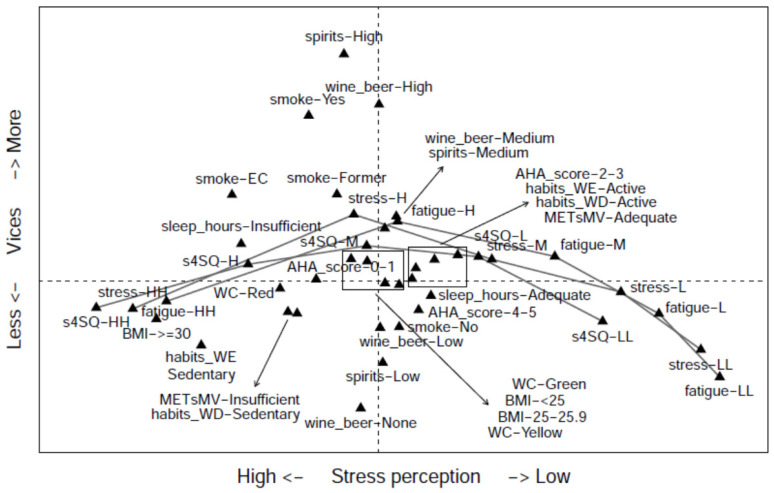
Multiple Correspondence Analysis map. The plot shows the modalities of the investigated variables (see the legend below) in the space spanned by the axes determined by MCA. The grey segments connect the modalities of the variables related to stress perception, namely s4SQ, stress, and fatigue. WC = waist circumference; AHA_Score = AHA nutrition score.; BMI = body mass index; METsMV = moderate and vigorous physical activity volume (adequate if ≥600 MET·min/week, otherwise insufficient); _WD = sedentary behaviour during working days (active if <9 h/week, otherwise sedentary); _WE = sedentary behaviour during weekends (active if <9 h/week, otherwise sedentary); sleep_hours = hours of sleep (adequate if ≥7 h/day, insufficient otherwise); stress: LL (low: 0–2), L (moderate/low: 3–4 points), M (moderate: 5–6 points), H (moderate/high = 7–8 points), HH (high: 9–10 points); fatigue: LL (low: 0–2), L (moderate/low: 3–4 points), M (moderate: 5–6 points), H (moderate/high = 7–8 points), HH (high: 9–10 points); s4SQ = short questionnaire on subjective somatic-stress-related symptoms: LL (low: 0–3 points), L (moderate/low: 4–6 points), M (moderate: 7–10 points), H (moderate/high: 11–15 points), HH (high: 16–30 points); smoke = smoking habits: SM (smoker), FSM (former smoker), EC (electronic cigarettes), and NS (non-smoker); Wine_beer = wine and beer consumption: none (0 glass/week), low, medium, high (0, 0.1–1, 1.1–2.9, and >2.9 glasses/week); spirits = spirit consumption: none (0 glasses/week), low (0.1–1 glasses/week), high (>1 glasses/week). The modalities of the variables of interest (e.g., smoker, non-smoker) are represented by points, and the presence of points close to one another reveals that the corresponding modalities are tendentially observed together. On the right side of the plot (*X*-axis), the following variables are found: fatigue = LL (very low), L (moderate/low), and M (moderate); stress = LL (very low), L (moderate/low), and M (moderate); and s4SQ = LL (very low) and L (moderate/low). Higher levels of the same variables are observed in the left part. Note that the different classes for perceptions of stress, fatigue, and somatic symptoms are in progressive order, as evidenced, respectively, by the lines. In the top part of the figure (*Y*-axis) and near each other, the following variables are found: smokers (also smokers of electronic cigarettes, represented by the label EC), former smokers, and the highest levels of wine and beer (represented by the label Wine_Beer-High) and spirit (Spirits-High) consumption. The lowest levels of the same variables are observed in the bottom part.

**Figure 2 nutrients-16-04339-f002:**
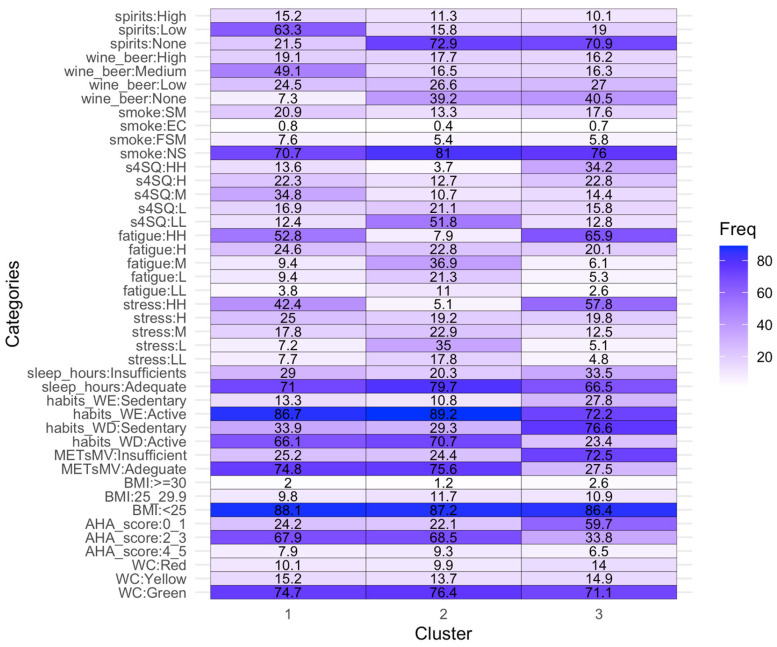
Heatmap showing distributions of the student’s features for each cluster. The *X*-axis represents the three clusters. The *Y*-axis represents all the possible modalities of the 13 variables used for cluster analysis. The numbers in the cells express the percentages of students, showing the modalities of the variables for each cluster. See the legend of Figure 1 for abbreviations.

**Figure 3 nutrients-16-04339-f003:**
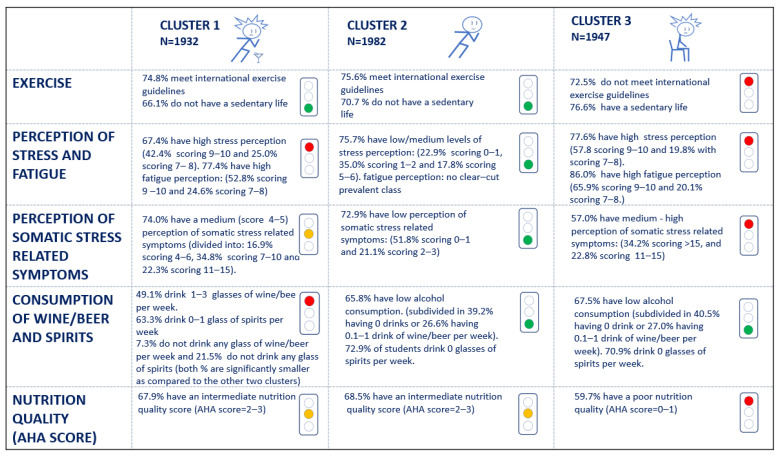
Characterization of students’ lifestyles according to the three clusters. AHA = American Heart Association.

**Figure 4 nutrients-16-04339-f004:**
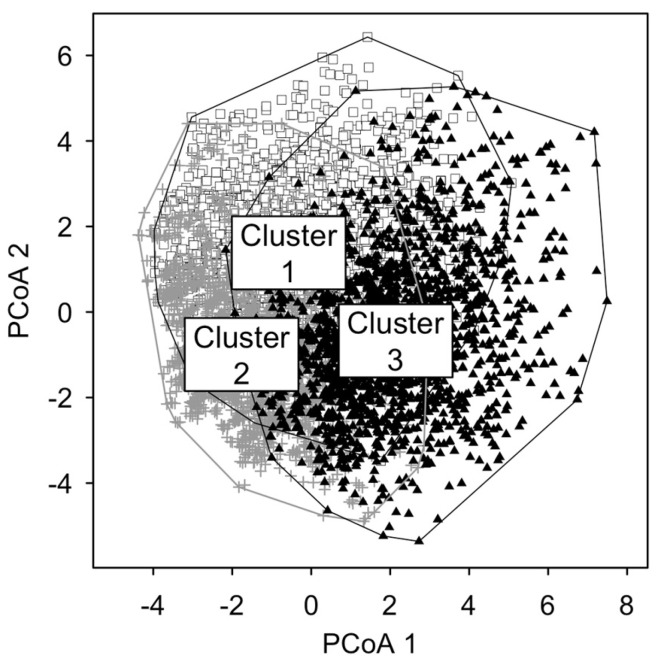
PCoA plot. The figure shows students represented in two-dimensional spaces that preserve the highest possible number of differences (goodness—of—fit index equal to 18.8%). Distinct labels represent students belonging to distinct clusters: empty squares for cluster 1, grey crosses for cluster 2, and black triangles for cluster 3.

**Table 1 nutrients-16-04339-t001:** Anthropometric and lifestyle data collected from all subjects (total) and divided by gender.

	Total (*n* = 6976)	Female (*n* = 3665)	Male (*n* = 3300)	*p*-Value
**Affiliation:**				
University of Milan	1782 (25.5%)	1221 (33.3%)	556 (16.8%)	
Polytechnic of Milan	5128 (73.5%)	2396 (65.4%)	2727 (82.5%)	
Other (not specified)	66 (0.9%)	48 (1.3%)	17 (0.5%)	
**Age [y]**	21 (20, 23)	21 (20, 23)	21 (20, 23)	ns
**Weight [Kg]**	63 (55, 72)	57 (52, 62)	71 (65, 78)	*p* < 0.0001
**Height [cm]**	172 (165, 179)	166 (161, 170)	179 (174, 183)	*p* < 0.0001
**BMI [Kg/m^2^]**	21.3 (19.6, 23.2)	20.5 (19.1, 22.3)	22.2 (20.6, 24.1)	*p* < 0.0001
Underweight/normal weight (<25 Kg/m^2^)	6125 (87.8%)	3353 (91.5%)	2762 (83.7%)	
Overweight (25–29.9 Kg/m^2^)	713 (10.2%)	258 (7.0%)	454 (13.8%)	
Obese (≥30 Kg/m^2^)	138 (2.0%)	54 (1.5%)	84 (2.5%)	
**Waist circumference [cm] ***	79 (70, 87)	71 (66, 80)	84 (80, 93)	*p* < 0.0001
Green	4342 (74.1%)	2378 (72.6%)	1964 (75.8%)	
Yellow	856 (14.6%)	467 (14.3%)	389 (15.0%)	
Red	664 (11.3%)	427 (13.1%)	237 (9.2%)	
**METsMV [MET·min/week]**	800 (200, 1800)	640 (120, 1440)	1200 (320, 2200)	*p* < 0.0001
Insufficient (<600 METs)	2845 (40.8%)	1736 (47.4%)	1104 (33.5%)	
Adequate (≥600 METs)	4131 (59.2%)	1929 (52.6%)	2196 (66.5%)	
**Frequency of strength exercise:**				*p* < 0.0001
Never	3719 (53.3%)	2137 (58.3%)	1577 (47.8%)	
Sometimes	889 (12.7%)	454 (12.4%)	434 (13.2%)	
1 session/week	580 (8.3%)	328 (8.9%)	249 (7.5%)	
2–3 sessions/week	1346 (19.3%)	619 (16.9%)	725 (22.0%)	
More than 3 sessions/week	442 (6.3%)	127 (3.5%)	315 (9.5%)	
**Frequency of flexibility exercise:**				*p* < 0.0001
Never	3271 (46.9%)	1574 (42.9%)	1695 (51.4%)	
Sometimes	1546 (22.2%)	900 (24.6%)	643 (19.5%)	
1 session/week	806 (11.6%)	495 (13.5%)	309 (9.4%)	
2–3 sessions/week	1003 (14.4%)	527 (14.4%)	474 (14.4%)	
More than 3 sessions/week	350 (5.0%)	169 (4.6%)	179 (5.4%)	
**Sedentary Behaviour:**				ns *p* < 0.0001
Week days [hours/week]	45 (35, 55)	45 (35, 55)	45 (35, 55)	
Weekend days [hours/week]	12 (10, 18)	12 (10, 16)	14 (10, 18)	
**AHA Diet Score [a.u.]**	2 (1, 3)	2 (1, 3)	2 (1, 2)	ns
**Smoking habits:**				ns
Non-smoker	5268 (75.5%)	2737 (74.7%)	2525 (76.5%)	
Ex-smoker	421 (6.0%)	227 (6.2%)	193 (5.8%)	
Electronic cigarette smoker	43 (0.6%)	21 (0.6%)	22 (0.7%)	
Smoker	1244 (17.8%)	680 (18.6%)	560 (17.0%)	
**Coffee [cups/day]:**				ns
0	2181 (31.3%)	1093 (29.8%)	1084 (32.8%)	
1–2	3495 (50.1%)	1881 (51.3%)	1611 (48.8%)	
3+	1300 (18.6%)	691 (18.8%)	605 (18.3%)	
**Wine and beer [glass/week]:**				*p* < 0.0001
0	2027 (29.1%)	1253 (34.2%)	772 (23.4%)	
>0–3	3675 (52.7%)	1943 (53%)	1727 (52.3%)	
4-	845 (12.1%)	336 (9.2%)	508 (15.4%)	
7+	428 (6.1%)	132 (3.6%)	293 (8.9%)	
**Spirits [glass/week]:**				*p* < 0.0001
0	3842 (55.1%)	2133 (58.2%)	1704 (51.6%)	
>0–3	2992 (42.9%)	1489 (40.6%)	1500 (45.5%)	
4–6	115 (1.6%)	32 (0.9%)	82 (2.5%)	
7+	27 (0.4%)	11 (0.3%)	14 (0.4%)	
**Short 4SQ score [au]**	8 (3, 13)	9 (5, 15)	6 (2, 11)	*p* < 0.0001
**Stress perception [au]**	6 (3, 8)	7 (4, 9)	5 (3, 8)	*p* < 0.0001
**Fatigue perception [au]**	7 (4, 9)	8 (5, 9)	6 (4, 8)	*p* < 0.0001
**Sleep [hours/night]**	7 (6, 8)	7 (6, 8)	7 (7, 8)	ns
**Perception of sleep quality [au]**	7 (6, 8)	7 (6, 8)	7 (6, 8)	ns
**Perception of health quality [au]**	7 (6, 8)	7 (5, 8)	7 (6, 8)	ns
**Perception of quality of life [au]**	7 (5, 8)	7 (5, 8)	7 (6, 8)	ns
**Presence of chronic disease**	630 (9.0%)	387 (10.6%)	241 (7.3%)	*p* < 0.0001
**Desire to be helped with lifestyle change**	5155 (73.9%)	2861 (78.1%)	2288 (69.3%)	*p* < 0.0001
**Desire to improve nutrition**	3677 (52.7%)	1864 (50.9%)	1807 (54.8%)	*p* = 0.0297
**Desire to improve exercise**	4652 (66.7%)	2617 (71.4%)	2030 (61.5%)	*p* < 0.0001
**Desire to improve stress management**	4098 (58.7%)	2380 (64.9%)	1715 (52.0%)	*p* < 0.0001

Data are presented as counts and proportions for categorical variables and medians and quartiles (Q1, Q3) for continuous ones. Eleven subjects declared to belong to the “other” gender; thus, they were accounted for only in the total group (column 2). * For waist circumference, results were calculated from data of 5862 students (3272 female, 2590 male) whose responses were considered reliable (see methods). Abbreviations: BMI = body mass index; AHA = American Heart Association; MET = Metabolic Equivalent; 4SQ = subjective somatic-stress-related symptoms questionnaire; au = arbitrary units; ns = not significant.

## Data Availability

Data will be available on justified request. We have not yet uploaded the data because they are part of an ongoing study on students’ lifestyles and we are preparing other papers using them.

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
