# Peer review of "Assessing Lifestyle in a Large Cohort of Undergraduate Students: Significance of Stress, Exercise and Nutrition"

_nutrients, 2024, doi:10.3390/nu16244339_

Round 1
Reviewer 1 Report
Comments and Suggestions for Authors
The present study aimed to define groups of students characterized by specific life-styles to better tailor preventive strategies and educational procedures to favour well-being and health during academic years. The manuscript is interesting and of value. However, it need a revision.
Please define the term lifestyle. What variables describe lifestyle in your study? Clearly say how you understand it and how it was measured. In the Introduction is not clearly defined.
Methods is well described and the analysis too. Discussion is informative.
Author Response
Comments and Suggestions for Authors
- The present study aimed to define groups of students characterized by specific life-styles to better tailor preventive strategies and educational procedures to favour well-being and health during academic years. The manuscript is interesting and of value. However, it need a revision. Please define the term lifestyle. What variables describe lifestyle in your study? Clearly say how you understand it and how it was measured. In the Introduction is not clearly defined. Methods is well described and the analysis too. Discussion is informative.
We thank the Reviewer for her/his kind comments. We added, as requested, the definition of “lifestyle” we used in the Introduction and better specified the variables we considered in the study. Moreover, considering also the Second Reviewer’s comments/requests, we inserted in the body of the methodology all the information regarding the lifestyle assessment and the measures employed to quantify exercise, nutrition, sleep, smoking, stress, etc.

Reviewer 2 Report
Comments and Suggestions for Authors
Methodology. Not all necessary information regarding the data included in the questionnaire was placed in this section. Some important information is also included in other sections of the manuscript (only the reference in sentence L159-161 to Appendix A), making it difficult to read the text and interpret the data. Why did the authors not decide to include the information in Appendix A in the body of the Methodology?
L150-157 and Table 1. Did gender differentiate lifestyle components in this research? It would be worth pointing out the differences between genders here.
L155-157. Were participants asked about specific types of support (e.g., workshops, individual counseling)?
Discussion Section. The lifestyle of students is shaped by various factors, including financial opportunities, access to health resources, and social norms that influence their approach to health, physical activity, nutrition, stimulants, and mental health. The manuscript lacks a discussion of the results with the results of studies by other researchers. Are the study results similar to previous studies on assessing the lifestyle of students from different countries, including Western Europe?
L228-238 and L416-429. What criteria were adopted for the diet assessment according to the AHA (American Heart Association) guidelines? Did the questionnaire include detailed questions about the amount and frequency of food consumption?
Author Response
Comments and Suggestions for Authors
- Not all necessary information regarding the data included in the questionnaire was placed in this section. Some important information is also included in other sections of the manuscript (only the reference in sentence L159-161 to Appendix A), making it difficult to read the text and interpret the data. Why did the authors not decide to include the information in Appendix A in the body of the Methodology?
We thank the Reviewer for this comment which can help to better understand the employed methodology. We decided to insert some information in the Appendix to simplify the method section. We agree that a longer, but easier to read, section is a better solution. We inserted in the body of the methodology all the information.
- L150-157 and Table 1. Did gender differentiate lifestyle components in this research? It would be worth pointing out the differences between genders here.
We added in Table 1 all one column inserting the p-value of the differences between males and females. We also added in the result section an in-depth description of Table 1, evidencing the main differences between males and females. Moreover, we add the percentage of males and females in the 3 evidenced student clusters.
- L155-157. Were participants asked about specific types of support (e.g., workshops, individual counseling)?
Participants were not asked about specific types of support. We agree with the Reviewer that this information would be important, nevertheless, we decided to avoid asking them too many questions: the list of possible specific types of support (considering exercise, nutrition, stress management, stopping smoking, etc) would be very long requiring a significant time to be completed. We will consider this issue in future assessments
- Discussion Section. The lifestyle of students is shaped by various factors, including financial opportunities, access to health resources, and social norms that influence their approach to health, physical activity, nutrition, stimulants, and mental health. The manuscript lacks a discussion of the results with the results of studies by other researchers. Are the study results similar to previous studies on assessing the lifestyle of students from different countries, including Western Europe?
We thank the Reviewer for this comment. We added in the discussion a paragraph dedicated to this issue.
- L228-238 and L416-429. What criteria were adopted for the diet assessment according to the AHA (American Heart Association) guidelines? Did the questionnaire include detailed questions about the amount and frequency of food consumption?
We employed the AHA diet score [26] (Circulation 2010, 121, 586–613, doi:10.1161/CIRCULATIONAHA.109.192703). it considers 5 simple questions related to the following items:
- Fruits and vegetables: ≥ 4.5 cups per day
- Fish: ≥two 3.5-oz servings per week (preferably oily fish)
- Fiber-rich whole grains (≥1.1 g of fiber per 10 g of carbohydrate): ≥three 1-oz-equivalent servings per day
- Sodium: <1500 mg per day
- Sugar-sweetened beverages: ≤ 450 kcal (36 oz) per week.”
Each item corresponds to 1 point. The sum of the points (from 0 to 5) defines the level of health:
- 0-1 points = poor health
- 2-3 points = intermediate health
- 4-5 points = ideal health
This score (as stated in the paper published [26]) is not intended to be comprehensive. Rather, it is a practical approach that provides individuals with a set of potential concrete actions. Moreover, it provides three categories (poor health, intermediate health and ideal health) which may be very useful to describe different subject groups.

Reviewer 3 Report
Comments and Suggestions for Authors
I believe that the authors of the article missed the opportunity to produce an article of international impact with a sample of more than 9,400 people. They performed a cluster analysis that, although it offered interesting information, did not have the impact that could have been had with another type of analysis (differences in means, more detailed multivariate studies, regression studies, prediction or mediation through structural equations).
This journal is a journal of international impact and the type of evidence published must have greater external validity or the possibility of generalization to other contexts.
This reviewer considers that the article requires in-depth changes, especially in the empirical approach and in its structure.
Abstract:
1. It would be necessary to include in the summary the main results obtained beyond reporting the % of the results of the cluster analysis.
Introduction:
2. A more in-depth review of the state of the art of the variables under study would be necessary. This entire section is resolved in less than one page.
3. It is necessary to include a reference to scientific works of international relevance, especially works from the last 5 years.
4. It would be necessary to include the objectives or hypotheses of the research at the end of the introduction section, in order to clearly visualize the purposes of the study.
Material and Method:
5. This reviewer suggests dividing this section into the following subsections: Participants, Instruments, Procedure and Data Analysis.
6. All the information is presented in a single section, which is not usual in research articles.
7. The reliability of the tests used is not reported; as I have already indicated, the absence of a subsection on instruments makes it difficult to visualize the characteristics of the information collection instruments.
8. Permissions and authorizations are obtained from the Ethics Committee of one of the authors' Universities. The authors have everything in their favor to present an article with much more evidence and more statistical analysis (inferential statistics).
9. The sample size is high, but an opportunity is lost to report on the type of sample used, as well as the size of the statistical power or the confidence level and sampling error of the subjects. Given the high sample size, these issues should be taken into account.
10. This reviewer considers that the data collected allow for the performance of other statistical tests that offer results that are more generalizable to other contexts, that is, results that allow the external validity of the study to be much greater.
11. A cluster analysis is carried out, which is reduced to its use to classify a group of individuals into homogeneous groups, that is, its objective is the classification of individuals.
12. This aim of the study is very poor, very conservative, the study with this sample size offers many more possibilities if other types of parametric tests and the use of inferential statistics had been carried out, since with the type of data collected, much more relevant evidence could be offered.
Results:
13. Different tests of differences in means based on gender are carried out in each of the variables considered, but only statistical significance (p value) is reported; but it is not reported in the appropriate manner. The value of the t test performed is not reported, nor is the effect size (Cohen's d) nor is the statistical power of the results. It only mentions and comments on the results of the cluster analysis.
Discussion:
14. The discussion would need to be presented based on the objectives or the hypotheses that would have been necessary to clearly state at the beginning of the study.
15. All these questions are not just formal questions.
16. In a scientific study it is not enough to include precise information; it is just as important to respect the structure and wording of the scientific article so that the manuscript can be interpreted and read by the scientific community.
17. The authors lose the opportunity to report results that are much more relevant to the community in general and to the scientific community in particular.
Conclusions:
18. The limitations of the study must be established, which are not specified in detail in the study.
19. Likewise, the future lines or prospects of the study should be considered.
20. The number of works cited in the study must be increased, especially in the international context, and especially from the last 3-4 years, to increase the index of topical relevance of the study.
Author Response
Reviewer 3
We thank this Reviewer for her/his comments which may help improve our paper. In the revised version now submitted, we mark in red all the changes inserted in our previous revision (revision 1) and in orange the changes asked by the third Reviewer (revision 2).
Please find here our answers to the third Reviewer’s specific requests.
COMMENT: I believe that the authors of the article missed the opportunity to produce an article of international impact with a sample of more than 9,400 people. They performed a cluster analysis that, although it offered interesting information, did not have the impact that could have been had with another type of analysis (differences in means, more detailed multivariate studies, regression studies, prediction or mediation through structural equations).This journal is a journal of international impact and the type of evidence published must have greater external validity or the possibility of generalization to other contexts. This reviewer considers that the article requires in-depth changes, especially in the empirical approach and in its structure.
RESPONSE: We thank the Reviewer for this comment, we are aware that our data can be analysed with different approaches in consideration of the large study population. These data are the result of a collaboration between two main academic institutions, they were collected to create tailored intervention programs to foster health and well-being. This paper was prepared with this aim in mind using an ad hoc statistical approach. We clearly define our goal in the Introduction: “The present study aimed to define groups of students characterized by specific lifestyles to better tailor preventive strategies and educational procedures to favour well-being and health during academic years.” We are analysing the data using other approaches with other aims, but considering the collaboration between the two academic institutions, before proceeding, we have to close the path forward to the publication of a paper with the first goal we settled together because we need to implement clinical intervention in our Universities to improve health, tailored to specific students’ characteristics.
Indeed, cluster analysis is an acknowledged tool to provide data segmentation and individualized profiling of subjects along the information available in the dataset. Parametric statistical inference should be the aim of next studies involving targeted study designs and sampling strategies stemming from the experience of the present work.
Therefore, we agree with the Reviewer on the importance of more detailed multivariate studies, regression studies, prediction or mediation through structural equations, but now we may not change the goal of the present study, we hope that this Reviewer will help us to proceed on this road.
Abstract:
COMMENT1. It would be necessary to include in the summary the main results obtained beyond reporting the % of the results of the cluster analysis.
RESPONSE: As requested, we better specify in the abstract the main results
Introduction:
COMMENT 2. A more in-depth review of the state of the art of the variables under study would be necessary. This entire section is resolved in less than one page.
COMMENT 3. It is necessary to include a reference to scientific works of international relevance, especially works from the last 5 years.
RESPONSE: As requested, we added in the introduction a reference to some works of international relevance, from the last 5 years, which addressed students’ lifestyles in different countries.
COMMENT 4. It would be necessary to include the objectives or hypotheses of the research at the end of the introduction section, in order to clearly visualize the purposes of the study.
RESPONSE: We improve the end of the introduction section, adding our hypothesis before stating the objective of the study
Material and Method:
COMMENT 5. This reviewer suggests dividing this section into the following subsections: Participants, Instruments, Procedure and Data Analysis.
COMMENT6. All the information is presented in a single section, which is not usual in research articles.
RESPONSE: As requested, we divided the Materials and Methods section into the suggested subsections
COMMENT 7. The reliability of the tests used is not reported; as I have already indicated, the absence of a subsection on instruments makes it difficult to visualize the characteristics of the information collection instruments.
RESPONSE: We improved a lot the presentation of the methodology employed to assess lifestyle also considering the other two Reviewers’ suggestions, adding in the main text all the information that we inserted in the appendix in the original version of our paper that this Reviewer had considered to make her/his suggestions. The employed lifestyle assessment methodology considers texts widely employed and validated (for instance IPAQ to assess physical activity, AHA diet score for nutrition) or simple scale to assess stress, symptoms, and fatigue perceptions that were already employed in many our other papers, that showed a strong correlation between these scales and objective parameters such as those derived from autonomic nervous system evaluation. (we quote in the paper some of these articles).
COMMENT 8. Permissions and authorizations are obtained from the Ethics Committee of one of the authors' Universities. The authors have everything in their favor to present an article with much more evidence and more statistical analysis (inferential statistics).
RESPONSE: The Ethical Committees of both Academic Institutions granted us permissions and authorizations to conduct the study. As stated in the answer to the initial reviewer’s comment, we will conduct more extended statistical analysis in future studies to in-depth analyze our data, nevertheless, the two Academic Institutions were interested in defining specific students’ characteristics and profiles useful to better tailor practical intervention and we have at this stage, to fulfil this requirement.
COMMENT 9. The sample size is high, but an opportunity is lost to report on the type of sample used, as well as the size of the statistical power or the confidence level and sampling error of the subjects. Given the high sample size, these issues should be considered.
RESPONSE: Since the nature of the study related to advanced description and profiling of a large dataset aspects related to the assessment of statistical power are less mandatory in the absence of primary hypotheses related to parametric inference. As stated by the Reviewer the sample size is high, and we were extracting major profiles which are expected to be very stable in the considered target population.
COMMENT 10. This reviewer considers that the data collected allow for the performance of other statistical tests that offer results that are more generalizable to other contexts, that is, results that allow the external validity of the study to be much greater.
RESPONSE: The external validity of the present study should be assessed through newly designed studies. We think the profiles provided very stable and valid in the considered context since the large sample size. However, considering the adopted sampling strategy, we can’t exclude bias phenomena, mainly on the prevalence of the identified profiles, which could be assessed in future validation studies also on a multicentric basis.
COMMENT 11. A cluster analysis is carried out, which is reduced to its use to classify a group of individuals into homogeneous groups, that is, its objective is the classification of individuals.
COMMENT 12. This aim of the study is very poor, very conservative, the study with this sample size offers many more possibilities if other types of parametric tests and the use of inferential statistics had been carried out, since with the type of data collected, much more relevant evidence could be offered.
RESPONSE: As already stated, we wanted to rely on simple but highly informative results providing novel information with respect to the past. We do not agree with the opinion that the aim of this first study is poor only because we were not adopting extended parametric inference. We will be taking into consideration the Reviewer suggestions for additional evaluation in future papers but at the moment we are happy to deploy the information related to the present results because we think it should be a robust message impacting the lifestyle policies for our Students.
Results:
COMMENT 13. Different tests of differences in means based on gender are carried out in each of the variables considered, but only statistical significance (p value) is reported; but it is not reported in the appropriate manner. The value of the t test performed is not reported, nor is the effect size (Cohen's d) nor is the statistical power of the results. It only mentions and comments on the results of the cluster analysis.
RESPONSE: The statistical tests were reported only to provide a measure of evidence. We provided information about the test used and the correction for multiplicity. We do not understand why the Reviewer is stating this is no the appropriate manner. The value of the t-test is not appropriate since we were not relying of this test, nor the Cohen's effect size and the statistical power is not a primary concern in the analyses provided according on gender.
Discussion:
COMMENT 14. The discussion would need to be presented based on the objectives or the hypotheses that would have been necessary to clearly state at the beginning of the study.
COMMENT 15. All these questions are not just formal questions.
COMMENT 16. In a scientific study it is not enough to include precise information; it is just as important to respect the structure and wording of the scientific article so that the manuscript can be interpreted and read by the scientific community.
COMMENT 17. The authors lose the opportunity to report results that are much more relevant to the community in general and to the scientific community in particular.
RESPONSE: We thank this Reviewer for this suggestion which helps to improve the text. We rewrite the initial sentence of the discussion, specifying the hypothesis and the aim. Moreover, we rewrite some other parts of the discussion adding a reference to some works of international relevance, from the last 5 years, which addressed students’ lifestyles in different countries.
Conclusions:
COMMENT 18. The limitations of the study must be established, which are not specified in detail in the study.
COMMENT 19. Likewise, the future lines or prospects of the study should be considered.
RESPONSEWe best enlarge the study limitation section by inserting the following sentence:
“Another limitation of our study should be acknowledged: we focus our attention only on the possibility of unveiling specific students’ clusters (using cluster analysis), possibly losing the opportunity to report other results relevant to the community which would require different statistical approaches. Our large dataset could have permitted in-depth analysis. Nevertheless, we decided to focus our attention only on a specific goal: to define groups of students characterized by specific lifestyles to better tailor interventions aimed at improving health and well-being, answering a precise question of our Academic Institutions. Future lines of data analysis will definitely consider different statistical approaches such as other types of parametric tests, the use of inferential statistics, etc. granting results that will be more generalizable to other contexts”
COMMENT 20. The number of works cited in the study must be increased, especially in the international context, and especially from the last 3-4 years, to increase the index of topical relevance of the study.
RESPONSE: We thank the Reviewer for her/his request. As compared to the original version of our paper we added 13 recent citations to increase the index of topical relevance of the study.
Round 2
Reviewer 2 Report
Comments and Suggestions for Authors
I am satisfied with the response to the comments. The authors have incorporated all comments into the manuscript.
I have no further comments.
Author Response
We thank this Reviewer for her/his kind comments